# *In Silico* Evaluation of Bioactive Compounds of *Artemisia pallens* Targeting the Efflux Protein of Multidrug-Resistant *Acinetobacter baumannii* (LAC-4 Strain)

**DOI:** 10.3390/molecules27165188

**Published:** 2022-08-15

**Authors:** Suvaiyarasan Suvaithenamudhan, Sivapunniyam Ananth, Vanitha Mariappan, Victor Violet Dhayabaran, Subbiah Parthasarathy, Pitchaipillai Sankar Ganesh, Esaki Muthu Shankar

**Affiliations:** 1Infection Biology, Department of Life Sciences, School of Life Sciences, Central University of Tamil Nadu, Thiruvarur 610 005, Tamil Nadu, India; 2Sivan Bioscience Research and Training Laboratory, Kumbakonam 612 401, Tamil Nadu, India; 3Center for Toxicology and Health Risk Studies, Faculty of Health Sciences, Universiti Kebangsaan Malaysia, Jalan Raja Muda Abdul Aziz, Kuala Lumpur 50300, Malaysia; 4Department of Biotechnology and Bioinformatics, Bishop Heber College, Tiruchirappalli 620 017, Tamil Nadu, India; 5Department of Bioinformatics, School of Life Sciences, Bharathidasan University, Tiruchirappalli 620 024, Tamil Nadu, India; 6Department of Microbiology, Saveetha Dental College and Hospital, Saveetha Institute of Medical and Technical Sciences, Chennai 600 077, Tamil Nadu, India

**Keywords:** *Acinetobacter baumannii*, *Artemisia pallens*, vulgarin, lilac alcohol, molecular docking, molecular dynamic simulation, efflux protein

## Abstract

*Acinetobacter baumannii* (*A. baumannii*) is one of the major representative aetiologies of recalcitrant nosocomial infections. Genotypic and phenotypic alterations in *A. baumannii* have resulted in a significant surge in multidrug resistance (MDR). Of all the factors responsible for the development of antimicrobial resistance (AMR), efflux protein pumps play a paramount role. In pursuit of a safe alternative for the prevention and control of *A. baumannii* infections, bioactive compounds from the aerial parts of the medicinal plant *Artemisia pallens* were studied. GC-MS analysis of the ethanol extract of *A. pallens* detected five major compounds: lilac alcohol A, spathulenol, lilac alcohol C, n-hexadecanoic acid, and vulgarin. In silico examinations were performed using the Schrödinger suite. Homology modelling was performed to predict the structure of the efflux protein of *A. baumannii*-LAC-4 strain (MDR Ab-EP). The identified bioactive compounds were analysed for their binding efficiency with MDR Ab-EP. High binding efficiency was observed with vulgarin with a glide score of −4.775 kcal/mol and stereoisomers of lilac alcohol A (−3.706 kcal/mol) and lilac alcohol C (−3.706 kcal/mol). Our molecular dynamic simulation studies unveiled the stability of the ligand–efflux protein complex. Vulgarin and lilac alcohol A possessed strong and stable binding efficiency with MDR Ab-EP. Furthermore, validation of the absorption, distribution, metabolism, excretion, and toxicity (ADMET) properties of the ligands strongly suggested that these compounds could serve as a lead molecule in the development of an alternate drug from *A. pallens*.

## 1. Introduction

*Acinetobacter baumannii* is an opportunistic Gram-negative bacterium implicated with nosocomial infections, viz. urinary tract infections, pneumonia, soft tissue and skin infections, meningitis, etc. [1]. *A. baumannii,* particularly in the hospital environment, has posed a serious threat to global well-being due to its concrete role in multidrug resistance (MDR) in the past two decades [2,3]. Nosocomial pneumonia caused by *A. baumannii* has a mortality rate of 30–75%, while community-acquired pneumonia (CAP) due to *A. baumannii* has been reported to have a mortality rate of 40–60% [4]. Recently, the SENTRY antimicrobial surveillance program reported a high prevalence in MDR *A. baumannii* across parts of India, Pakistan, Malaysia, Thailand, and Taiwan [5,6,7]. The phenotypic and genotypic alterations in *A. baumannii* have resulted in an upsurge in antimicrobial resistance and virulence attributes [8,9]. The outer-membrane porins (OMPs), capsular polysaccharides, acinetobactin transporters, and K1 surface antigen protein 1 are some of the key factors responsible for promoting *A. baumannii* as a notorious nosocomial pathogen [10,11]. *A. baumannii* holds a plethora of virulence genes that encode for biofilm formation, facilitate adherence to different surfaces [12], maintain viability at extreme conditions, and protect against antimicrobial agents as well as the immune surveillance radars of the host [13,14]. Conventional antibiotics such as cephalosporins, penicillin, tetracycline, quinolones, and aminoglycosides have often been proven to be ineffective against *A. baumannii* owing to the resistance determinants of the bacterium. The underlying resistance mechanism in genetic alterations appears to cause configurational changes in membrane fusion proteins, overexpression of antimicrobial enzymes and efflux transporters, alterations in target sites, and insertion of novel resistance determinants [15]. The key factors implicated in imparting resistance in *A. baumannii* include β-lactamases (enzymatic mechanism), efflux protein pumps and membrane permeability (nonenzymatic mechanism), and alterations in penicillin-binding protein (PBP) sequences [16,17,18]. The overexpression of efflux proteins in the bacterial cell membrane is a critical factor that offers multidrug resistance in *A. baumannii* [19]. The high-level resistance developed by *A. baumannii* against conventional antimicrobial drugs has made its control challenging and cumbersome, warranting the need for an effective antimicrobial agent.

Given the surging quantum of adverse events, modern chemical and synthetic drugs have lost allure in the general public, and, hence, traditional medicines utilizing natural products derived predominantly from plant sources are gaining importance due to their eco-friendly attributes and negligible sideeffects. Medicinal plants are unique in their potential to cure a broad spectrum of diseases owing to the presence of phytochemicals that exhibit diverse pharmacological properties. Hence, plants represent the appropriate candidates in our search and discovery of newer drug compounds [20]. *Artemisia pallens* is one such highly valuable medicinal plant belonging to the family Asteraceae [21].It is popularly known as “Davana” in Ayurveda [22] and is traditionally used in the treatment of diabetes, hypertension, and depression [23]. The presence of a plethora of secondary metabolites renders this plant an ideal candidate for use as a herbal therapeutic agent against various ailments.The essential oil present in *A. pallens* is a flavouring agent [24], and is well-known for its antibacterial, antifungal, antispasmodic [25], antihelmintic [26], and antioxidant activities [27].

The screening of phytochemicals in medicinal plants aids in determining the pharmacological activities. Chromatography (GC-MS) and spectroscopic Fourier-transform infrared spectroscopy (FTIR) analysis are routinely used to identify functional groups and bioactive compounds in plants. Furthermore, sophisticated software is used in drug discovery to screen drugs from phytochemicals [26]. Computational prediction tools ease the in silico prediction of pharmacological, pharmacokinetic, and toxicological performances [27]. Molecular docking is a highly successful and low-cost method for designing and validating pharmaceuticals, and provides critical information on drug—receptor interactions to anticipate the binding of ligands (drug) with target proteins. In silico studies on the inhibition of AdeABC integral protein of the efflux pump [28] and RND efflux pump protein in *A. baumannii* have previously been reported [29]. Here, we evaluated the bioactive compounds isolated from the aerial parts of *A*. *pallens* for their antibacterial efficacy against MDR *A. baumannii* via in silico (molecular docking and dynamic simulations studies) approaches. The interaction pattern of bioactive compounds with multidrug resistance efflux protein (MDR Ab-EP) complexes were predicted and validated for the designing of newer drug molecules.

## 2. Result and Discussion

The phytochemical constituents of *A. pallens* present in the ethanol extract were identified using GC-MS analysis (Figure 1), which revealed the presence of twenty-five different bioactive compounds. The major compounds lilac alcohol C (32.77%), spathulenol (3.03%), lilac alcohol A (7.29%), n-hexadecanoic acid (7.92%) and vulgarin (15.14%) were tabulated with their retention times, molecular formulae, and molecular weights. The stereoisomers of lilac alcohol occupied a total of 40% of the identified bioactive compounds (Table 1). *Syringa vulgaris* flower extract exhibited the presence of lilac alcohol stereoisomers C and D, which occupied >40% of the identified bioactive compounds [30]. Vulgarin is a sesquiterpene reported in *Artemisia judaica* possessing antihyperlipidemic and antihyperglycemic activities [31]. *Halymeniad urvillei* extract contains hexadecenoic acid [32]. The n-hexadecanoic acid recovered from aerial parts of *Rhanterium epapposum* appears to possess antibacterial activities [33]. Similarly, Doughari and Saa-Aondo [34] have reported that n-hexadecanoic acid isolated from the methanol extract of *Prosopis africana* was effective against *Klebsiella pneumoniae*, *Pseudomonas aeruginosa,* and *Microsporum canis* isolates. Moreover, spathulenol present in the n-hexane extract of *Ocotea notata* leaves showed inhibitory activity against *Mycobacterium bovis*.

### 2.1. ADMET Analysis

To be able to consider a bioactive compound as a lead molecule in the drug discovery process, the evaluation of its ADMET properties is essential. The druglikeness attributes of the identified bioactive compounds of *A. pallens* were evaluated for their ADMET characteristics using the Schrodinger software ‘QikProp’ module. Detailed analyses of QPlogPo/w (octanol/water coefficient), percent humanoral absorption, QPlogBB (brain/blood coefficient) and QPlogS (solubility) were performed and presented (Table 2). The drug likeness of ligands based on the Lipinski rule of 5 is vital in the rational designing of a drug. The percentage of human oral absorption of lilac alcohol A, C, and spathulenol was found to be 100%. The ligands n-hexadecanoic acid and vulgarin measured 87.13% and 83.57% QP, respectively. All the selected ligands possessed excellent absorption properties. The permissible range of solvent accessible surface area (SASA) for ligands ranged between 300 and 1000. The ligands evaluated for toxicity prediction showed admissible range of SASA ranging between 420 and 675 indicating the druggable nature of the ligands. Both the stereoisomers of lilac alcohol (A and C) exhibited SASA values of 420.285. The ligands vulgarin and spathulenol recorded a SASA value of 479.884 and 451.722, respectively. A high SASA value was observed with n-hexadecanoic acid (675.898). The hydrogen bond donors of the ligand molecules were 1, and hydrogen bond acceptors ranged between 0.75 and 5.75. Spathulenol had minimum hydrogen acceptors while vulgarin exhibited maximum hydrogen bond acceptors. The octanol/water coefficient was minimum for vulgarin (1.383) and maximum for n-hexadecanoic acid (5.282).Similarly, n-hexadecanoic acid showed a low blood/brain coefficient (−1.494) and solubility (−5.593). The evaluated ligands showed high percentage of absorption and low solubility (hydrophobicity). Together, we concluded that all the selected ligands showed no violation of the rule of Lipinski in order to be considered as drugs.

### 2.2. Homology Modelling and Validation of Predicted 3D Structure of A. baumanni Efflux Protein

In the present study, the 3D structure of a small multidrug-resistant transporter protein of *E. coli* (7JK8-A), that showed 96% identity with efflux protein of *A. baumannii*-LAC 4 strain, was used as a template to construct a 3D structure of MDR Ab-EP.The 3D structure and Ramachandran plots are presented in Figure 2. The efflux protein of *A. baumanni* had 84 amino acids. The stereochemical properties of the modelled efflux protein MDR Ab-EP were evaluated with a Ramachandran plot to understand the conformations of the MDR Ab-EP structure. The dihedral angles phi (Φ) and psi (ψ) of amino acid residues predicting the allowed and disallowed conformations in protein structure were analysed with Ramachandran plot. The plot showed that MDR Ab-EP protein had 94.4% of residues in the most favoured region and 5.6% residues in the allowed region.

### 2.3. Validation of Docking Process

The preciseness of the docking process depends on the prediction of binding pose by the docking software. The Glide XP docking was validated with the binding of selected ligands to the active sites of the target protein (*A. baumannii* efflux protein-MDR Ab-EP). Two binding sites were identified in MDR Ab-EP (Figure 3). The Site scores(S) of binding sites 1 and 2 were 0.844 and 0.742, respectively, and the drug ability score (D) was observed as 0.861 and 0.66, respectively. The binding site with the maximum D score (Site-1) was selected for further in silico studies (Table 3).

### 2.4. Molecular Docking

The major bioactive compounds identified in *A. pallens,* including lilac alcohol C (033081-34-4), spathulenol (006750-60-3), lilac alcohol A (033081-36-6), n-hexadecanoic acid (000057-10-3), and vulgarin (000148-21-7), were investigated for their binding efficiencies with *A. baumannii* efflux protein (MDR Ab-EP), and their 2D structures arepresented in Figure 4. The interacting amino acids, H-bonded interaction, bond length, Glide score (Kcal/mol), and MM-GBSA ΔG_bind_ (kcal/mol) of the protein–ligand complexes are shown in Table 4. The ligands lilac alcohol A and lilac alcohol C bound at binding site 1 of the target protein exhibited a Glide score of −3.706 (kcal/mol) and binding energy (ΔG_bind_) of −24.54 kcal/mol. Hydrogen bond interactions were observed with the OH group of TYR3 and LEU6 amino acid residues with a bond length of 1.74 Å and 2.17 Å, respectively. The interaction of the ligands lilac alcohol A and lilac alcohol C and the modelled efflux protein is presented in Figure 5A,C. The docking of spathulenol with the efflux protein was mediated with the OH group of TYR3 residue with a bond length of 2.08 Å, Glide score of −3.652 (kcal/mol), and binding energy (ΔG_bind_) of −30.51 kcal/mol. The docking of n-hexadecanoic acid with the efflux protein exhibited a Glide score of −3.706 kcal/mol, binding energy (ΔG_bind_) of −33.19 kcal/mol, and hydrogen bond interactions were observed with the OH group of ASN101 amino acid residue with a bond length of 1.85 Å. The ligand vulgarin showed two interactions with the TYR3 residue of the efflux protein through OH groups with bond lengths 2.05 Å and 2.13 Å. The Glide score was found to be −4.775 kcal/mol and the binding energy (ΔG_bind_) was−39.34 kcal/mol, which is the minimum score observed among the selected ligand molecules (Table 4 and Figure 5). The lowest Glide score and binding energy indicate that vulgarin is a potential drug candidate with high affinity towards the efflux protein of *A. baumannii*. From the molecular-docking simulation, it could be inferred that three compounds, namely vulgarin, spathulenol, and lilac alcohol C, can be considered as potent efflux protein inhibitor candidates, and can be used as potential antibacterial drugs.

### 2.5. Molecular Dynamics

Molecular simulation studies were performed using GROMACS. The root mean square deviation (RMSD) and root mean square fluctuation (RMSF) values determine the stability of the protein backbone and the ligand complexes. The RMSD plot (Figure 6A) presents the variations in the residues of the efflux protein backbone. Initial variations in residues were observed until11 ns, after which equilibration of the backbone structure was attained untilthe completion of the simulation run (50 ns). The high peak represented by the RMSF plot (Figure 6B) between 20 and 29 residues and between 80 and 86 residues indicates the formation of a loop configuration. The molecular dynamic simulation of MDR Ab-EP with ligand complexes was performed to evaluate stability (Figure 7). The comparison of protein–ligand complexes suggests that MDR Ab-EP and ligand (vulgarin) complex was highly stable. The stability of the protein–ligand complex was observed from the start of the run and further deviation was not observed untilthe end of the run (50 ns). The complex consisting of lilac alcohol C showed initial variation until9 ns and was found to be stabilized up to 50 ns simulation time (Figure 7A). Likewise, the spathulenol-MDR Ab-EP complex’s RMSD plot showed stability from 11 ns to 50 ns. Figure 7 represents the mean variation plot of the selected ligand–efflux protein complexes, which showed that the vulgarin–efflux protein complex (see Figure 7B) showed the least variation (0.29). The average hydrogen bond interactions values are shown in Table 5.

The fluctuations in the RMSF plot exhibited with high peaks indicate that the residues present in the backbone of the protein are involved in loop formation. This high deviation in theRMSD and RMSF plot observed for ligand n-hexadecanoic acid and lilac alcohol A suggests that the complexes were not stable until10 ns of the simulation run. From the plots of RMSF, it could be inferred that the protein–ligand complex involving n-hexadecanoic acid is the least stable compared to other ligands evaluated (Figure 7C). Two H-bond interactions were observed constantly in efflux protein–vulgarin (TYR3and TYR3), and lilac alcohol A, C (TRY3 and LEU6), indicating the strong and persistent binding of the ligand to the active site of the efflux protein. The average hydrogen bond interactions formed between the MDR Ab-EP along with the five selected ligand molecules are presented in Appendix A, respectively. The average hydrogen bond interaction values are given in Table 5. The MDR Ab-EP interaction with ligand 000148-21-7 (vulgarin) showed the highest average hydrogen bonding of 3.04.

## 3. Materials and Methods

*A. pallens* was collected from Kolli Hills, Tamil Nadu, India, and was authenticated (No. 3117) by a competent taxonomist from the Rapinat Herbarium, St. Joseph’s College (Autonomous), Tiruchirappalli, Tamil Nadu, India. The plant was washed with distilled water thrice and transferred to the processing laboratory using polyethene bags. The aerial parts of the plant were cut into pieces, washed, and shadow-dried for 10 days. Subsequently, the plant parts were ground to a fine powder using a mechanical blender. The dried powder was stored at 4 °C until further experiments.

### 3.1. Preparation of Solvent Extract

Crude plant extract was prepared by the Soxhlet extraction method. Briefly, 20 g of powdered plant material was uniformly packed into a thimble and extracted with 250 mL of ethanol. Extraction was continued for 24 h or untilthe solvent in the siphon tube of the extractor became colourless. The extract was collected in a beaker and condensed on a rotary evaporator under reduced pressure. Crude extract was stored at 4 °C for future use [35].

### 3.2. Identification of Bioactive Compounds from Aerial Parts of A. pallens

The phytochemical composition of ethanolic extract of *A. pallens* was analysed using a GC-MS (Agilent GC 7890A/MS5975C) chromatograph equipped with a Shimadzu QP-500 mass spectrometer, as per standard protocols [36]. The plant extract was dissolved in ethanol (1:25 ratio), and 1μL of the sample was subjected to analysis. A fused-silica column coated with polydimethyl siloxane (30 m length/0.25 mm diameter/0.25 μm film thickness) was used as the stationary phase. Helium was used as carrier gas at a flow rate of 1ml/min. The temperature of the injector was fixed at 325 °C. The temperature of the oven was maintained at 50 °C for a min, which was then increased to 300 °C at arate of 12 °C/min. An ionization voltage of 70 eV with 1:5 split rate was maintained. The composition of the extract was identified by comparing the mass spectra with known compounds (or) published data and the results are given in Appendix A.

### 3.3. Hardware and Software

The in silico analysis was performed on IBM Rack server (X3550M4 1U) with a dual Intel Xeon (E5-2670V210C) 2.5 GHz processor operated with the Linux operating system(Cent OS V6.5). The protein structure prediction, molecular docking, and dynamic simulation studies were performed with Schrodinger suite (2022-1) with graphical interface (Maestro 22-1) [37] and GROMACS package (version 4.5).

### 3.4. Ligand-Based Pharmacokinetics Analysis

In order to consider a compound to be a lead molecule in the drug discovery process, the ADMET properties arecrucial. The druglikeness parameters of the identified ligands from *A. pallens* were evaluated for ADMET characteristics using the Schrödinger software’s ‘QikProp’ module [38]. Detailed analyses of logP (octanol/water), QP% (human oral adsorption), QPlogBB (blood–brain barrier), and QPlogS (aqueous solubility) were conducted. The drug likeness of the ligands based on the Lipinski rule of 5 is vital in the rational designing of a drug.

### 3.5. Homology Modelling and Protein Structure Prediction

Homology modelling of the MDR efflux protein of *A. baumannii* (LAC-4) was constructed using a ‘Prime module’ of the Schrödinger software using the 3D structure of a small multidrug-resistant (SMR) transporter protein of *Escherichia coli* (7JK8-A) as a template as the protein showed 96% similarity [39]. The sequence of the efflux protein of the *A. baumannii* LAC-4 strain with the accession number AIY36556.1was retrieved from the NCBI database [40] in FASTA format and was used for predicting the 3D structure of the *A. baumannii* efflux protein. Loops were refined and verified using Schrödinger ‘Protein Refinement module’.

### 3.6. Preparation of Protein and Ligand Molecules

The assignment of bond orders, addition of hydrogen, and removal of water molecules were carried out with Schrödinger ‘Protein preparation wizard’ for the preparation and optimization of the efflux protein 3D structure. Protein protonation at biological pH was attained with the ‘Epik module’ of the Schrödinger software suite [41,42]. The hydrogen-bonding network was assigned using OPLS3e force field [43] and minimization was set to terminate when RMSD reached maximum cut-off (0.30 Å) to obtain the least possible energy [44].

The 2D structure of the ligands (bioactive compounds) identified from the aerial parts of *A. pallens* was retrieved from the National Institute Standard and Technology (NIST) Chemistry Web Book [45]. The 3D and geometry optimizations with energy minimization of ligands were executed using algorithms monitored in Schrödinger Maestro (v 22-1). The ‘LigPrep’ module [46] was used from the Maestro builder panel to prepare ligand and to generate 3D structure of the ligands by adding hydrogen atoms and removing salt and ionizing at pH (7 ± 2) using “Epik module” [47]. Energy minimization was performed using OPLS4 force field [48] by using the standard energy function of molecular mechanics and RMSD cut off 0.01 Å to generate the low-energy ring confirmation per ligand [49].

### 3.7. Active Site Prediction

The active sites were predicted with the ‘SiteMap’ [50,51] module based on the physical and structural properties of the modelled protein (MDR Ab-EP). The possible and potential active site was determined with qualitative site-score values considering the size, enclosure, contacts, hydrophilicity, hydrophobicity, and the balance between hydrogen donor and acceptor.

### 3.8. Molecular Docking

The bioactive compounds (ligands) identified from the aerial parts of *A. pallens* including lilac alcohol A (CAS No. 033081-34-4), spathulenol (CAS No. 006750-60-3), lilac alcohol C (CAS No. 033081-36-6), n-hexadecanoic acid (CAS No. 000057-10-3), and vulgarin (CAS No. 000148-21-7) were examined for their putative antibacterial activities against MDR-*A. baumannii* through molecular-docking analysis using the ‘Glide XP module’ of the Schrodinger software suite. The 3D structures of *A. pallens* phytocompounds were optimized for the docking conformation study. The structure-based molecular docking was performed to determine the efflux protein (MDR Ab-EP)—ligand interaction. Docking was carried out after constructing the grid by selecting the amino acid residues present in the binding site within the radius of 3 Å as the centroid. Default parameters were selected by keeping the ligands flexible on the docking calculation set. The formation of hydrogen bonds between the ligands and the residues of the active site, its length, and Glide XP (extra-precision) score were recorded [52,53,54].

### 3.9. Molecular Dynamics (MD) Simulations Analysis

Molecular dynamics simulations of efflux protein–ligand complexes were performed using GromacsV-5.1.4 package. The stability of the MDR Ab-EP and selected ligands was assessed. The automatic topology builder in the GROMOS96 53a6 force field for protein–ligand complexes [55,56] was used to construct the topology files for the proteins. PRODRG server [57] was used to construct the ligand topologies. The compounds were placed in a cubic box containing water molecules with a simple point charge (SPC). By adding the necessary counter ions, Na^+^ and Cl^−^, the system’s net charge was neutralized. The particle mesh Ewald (PME) approach was used to calculate the long-range electrostatic interactions. The linear constraint solver (LINCS) technique was used to constrain bond lengths using hydrogen atoms [58]. The equilibration of the protein–ligand complexes was conducted in two phases: the first conducted under an NVT ensemble (constant temperature and constant volume) for 100 ps to equilibrate the system with proteins and ligands at constant volume and temperature (300 K); and the second phase under an NPT (constant temperature and constant pressure) ensemble, wherein the number of particles, pressure, and temperature were kept constant. NPT was also used for 100 ps to equilibrate the system with proteins and ligands at constant volume, pressure (1 bar), and temperature (300 K). After completion of the two equilibration phases, the systems were well-equilibrated at the desired temperature and pressure for MD analyses. For each ligand with modelled structures of the MDR-EP Ab-LAC-4 strain, the final MD run was set at 50 ns. The trajectories generated by MD simulations were preserved and analysed further with GROMACS analytic tools [59,60].

## 4. Conclusions

The in silico evaluation carried out in the current work demonstrated the binding activity of bioactive compounds of *A. pallens* with the efflux protein of *A. baumannii*. The ADMET prediction for the selected ligands could help in identifying the druglikeness property for them to be used as lead molecules in drug discovery. The stability of the ligands in binding with the target efflux protein of *A. baumannii* was assessed with molecular dynamic simulation investigations. From our in silico evaluation, vulgarin and lilac alcohol A were identified as potent ligands with strong and stable binding potentials with the target protein MDR Ab-EP. The antimicrobial activity of the reported compounds should be validated through qualitative estimations (such as agar diffusion assay and minimal inhibitory concentration (MIC)) for future consideration as antibacterial agents. Taken together, the two ligands can be considered as potent lead molecules against the MDR-*A. baumannii* LAC-4 strain.

## Figures and Tables

**Figure 1 molecules-27-05188-f001:**
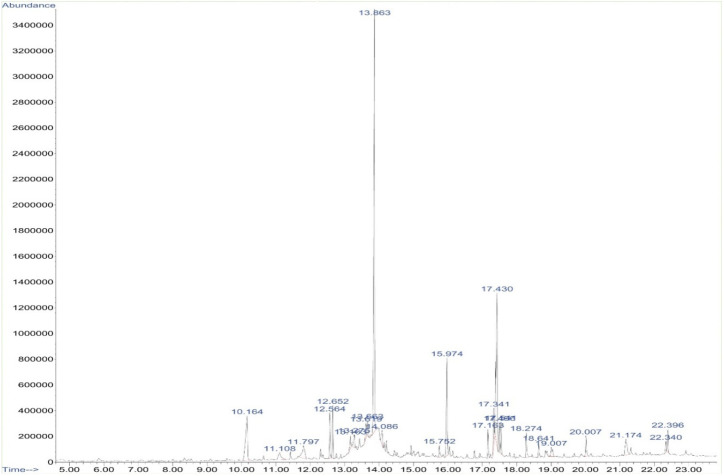
GC-MS chromatogram of ethanolic extract of *Artemisia pallans*. *X*-axis depicts time (in minutes) and *Y*-axis indicates abundance (in millivolts).

**Figure 2 molecules-27-05188-f002:**
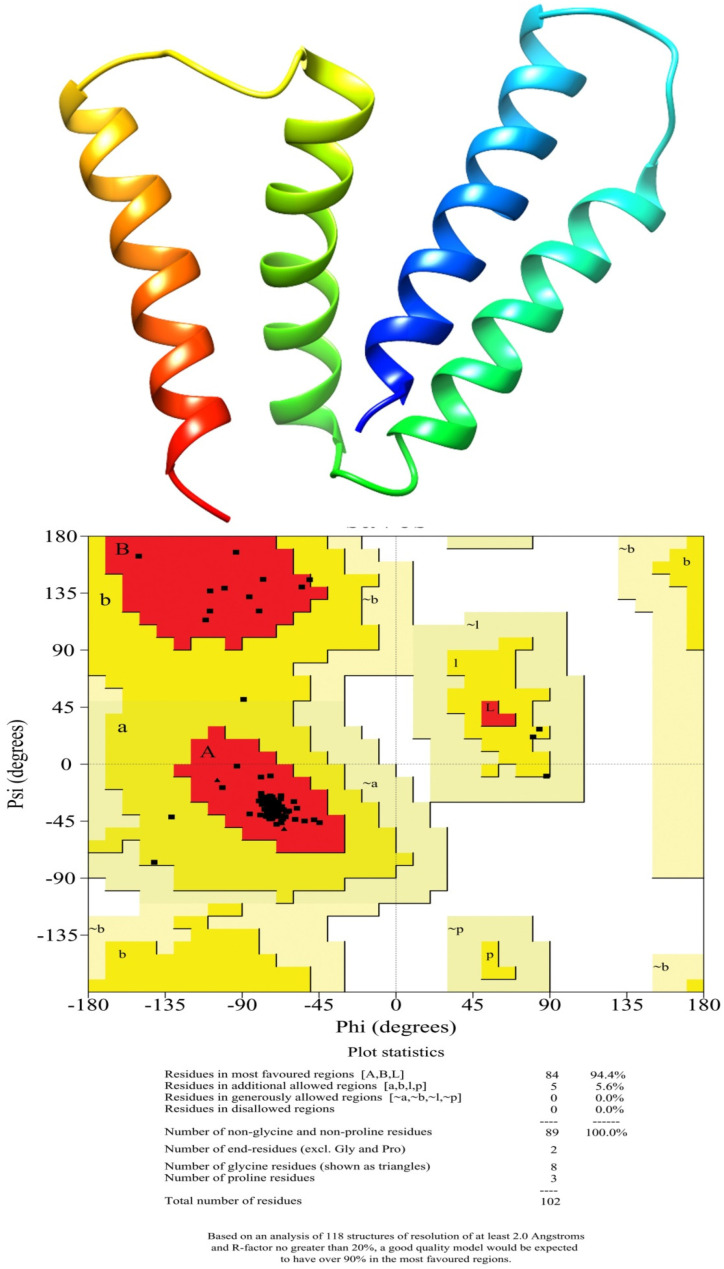
Homology model of the MDR efflux protein of *A. baumannii* (LAC-4) and Ramachandran plot analysis of modelled protein.

**Figure 3 molecules-27-05188-f003:**
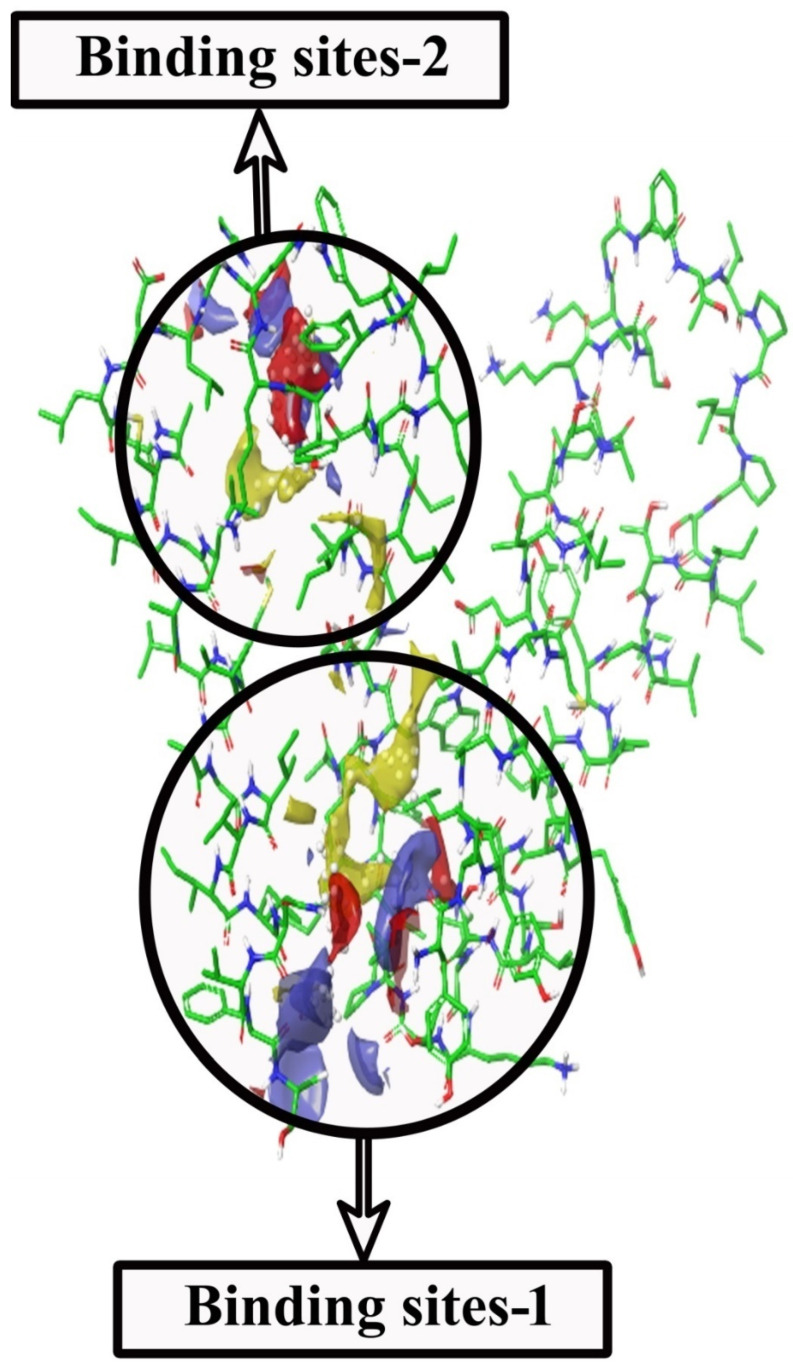
SiteMap shows structural complementarities of the efflux protein of *A. baumannii* (LAC-4) structure. (i) Red colour for H-bond acceptor sites (ii) Blue for H-bond donor sites (iii) Yellow for hydrophobic sites.

**Figure 4 molecules-27-05188-f004:**
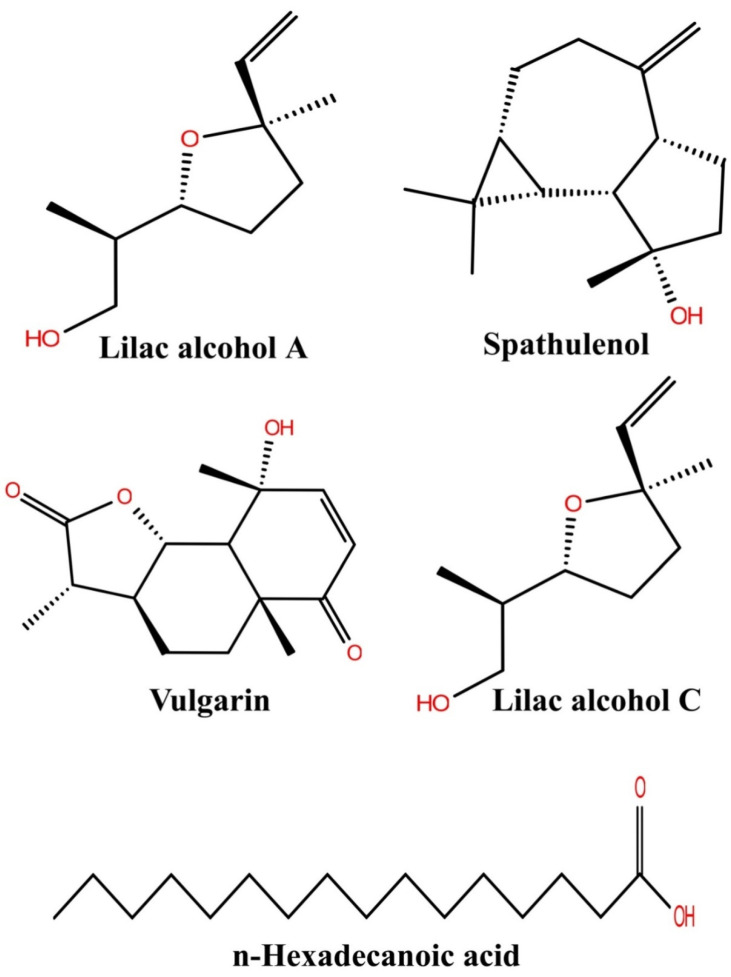
Two-dimensional structure of the compound isolated from the *Artemisia pallans*.

**Figure 5 molecules-27-05188-f005:**
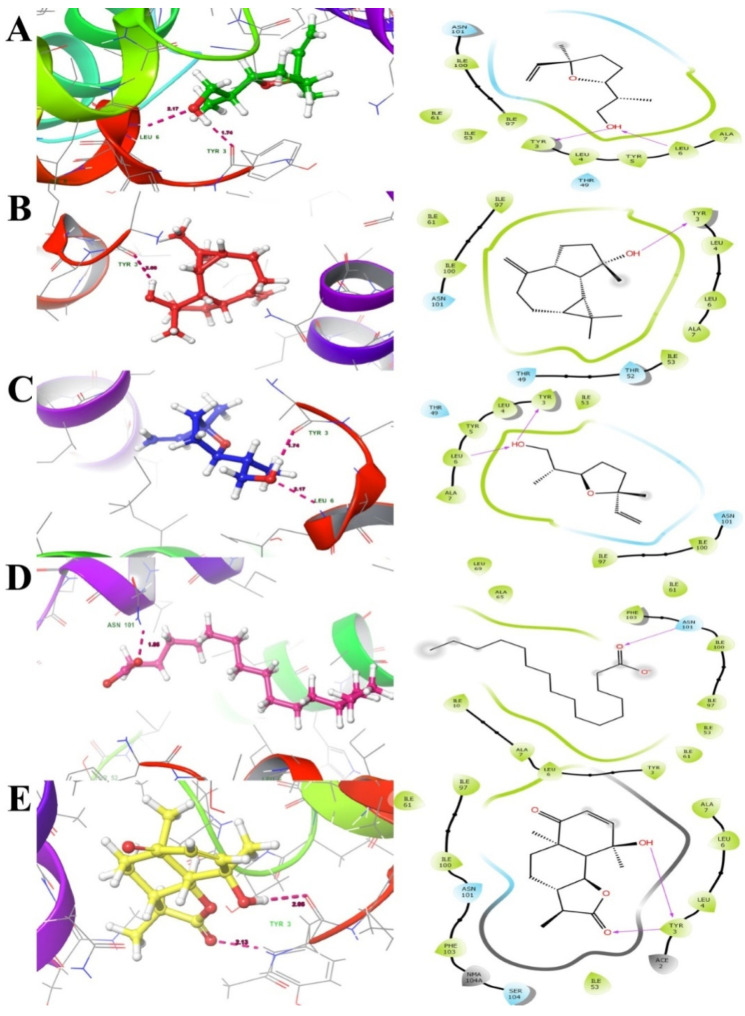
Docking interactions of ligands ((**A**). 033081-34-4, (**B**). 006750-60-3, (**C**). 033081-36-6, (**D**). 000057-10-3 and (**E**). 000148-21-7) with efflux protein ligand docking poses represented in threedimensions (Left) and twodimensions (Right). The ligands are shown as a ball and stick model and the dotted pink lines indicate hydrogen bond interactions and the residues.

**Figure 6 molecules-27-05188-f006:**
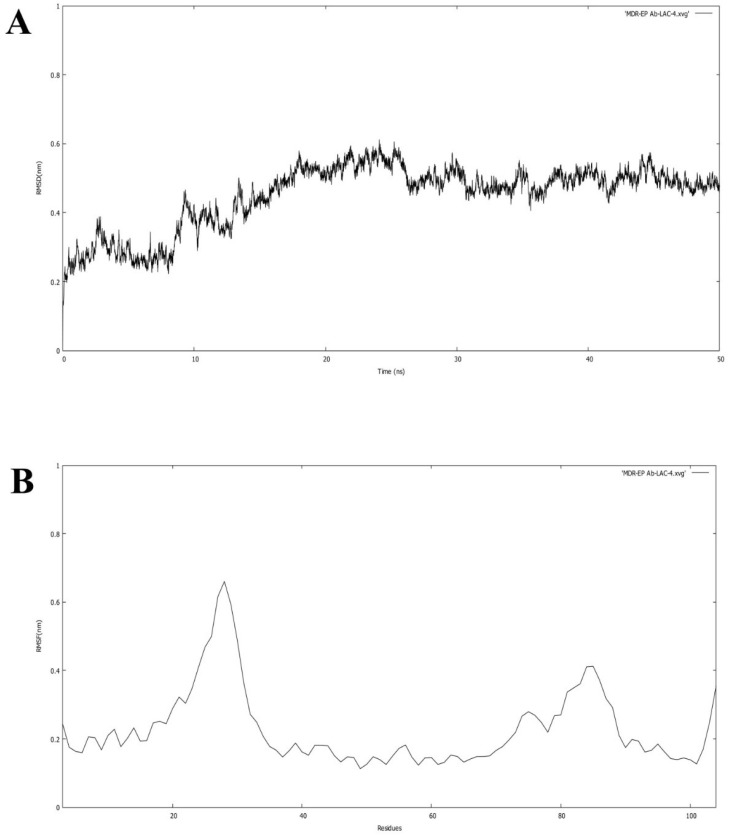
(**A**) RMSD of the backbone of efflux protein forms during the simulation run of 50 ns. (**B**) RMSF of the protein for each residue of the efflux protein.

**Figure 7 molecules-27-05188-f007:**
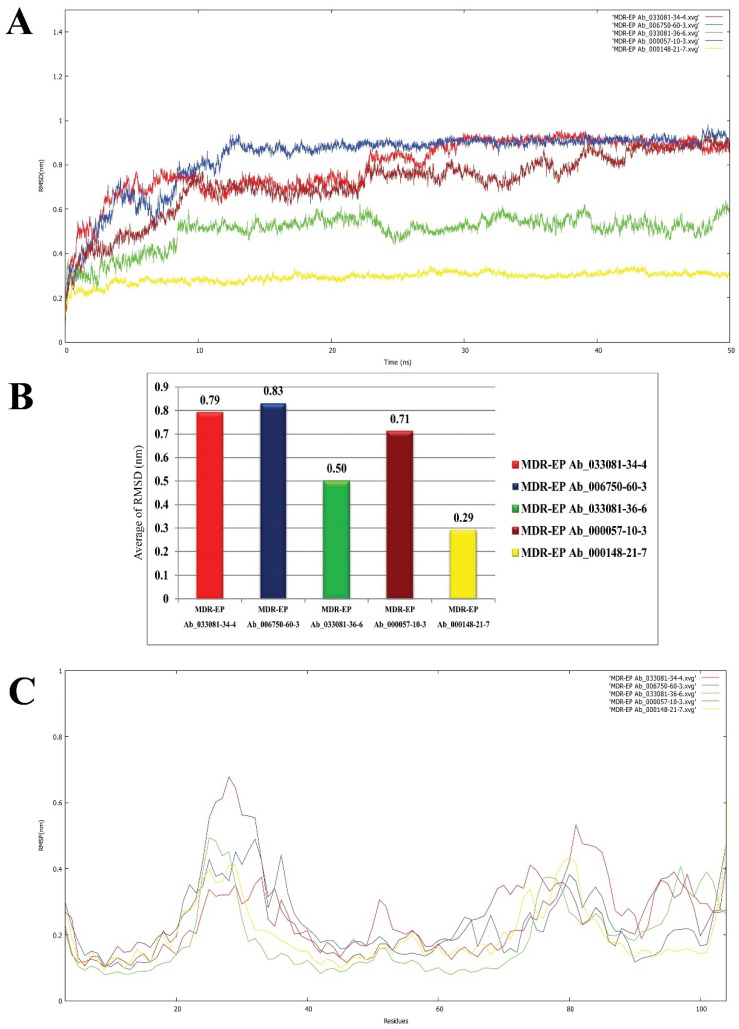
Backbone RMSD values of ligand nominees for efflux protein. (**A**) Red indicates MDR Ab-EP-033081-34-4 complex, blue indicates MDR Ab-EP-006750-60-3 complex, green indicates MDR Ab-EP -033081-36-6 complex, brown indicates MDR Ab-EP-000057-10-3, and yellow MDR Ab-EP-000148-21-7complex. (**B**) Bar diagram indicating the average of RMSD values for the protein–ligand complexes. (**C**) RMSF value of protein–ligand complexes during the trajectory period of simulation.

**Table 1 molecules-27-05188-t001:** Selected bioactive compounds of *A. pallens* (GC-MS analysis).

Sl. No	Retention Time	CAS Registry Number	Name of the Compound	Molecular Formula	Molecular Weight	Peak Area (%)
1	RT-10.164	033081-34-4	Lilac alcohol A	C_10_H_18_O_2_	170.251	7.29
2	RT-12.652	006750-60-3	Spathulenol	C_15_H_24_O	220.354	3.03
3	RT-13.863	033081-36-6	Lilac alcohol C	C_10_H_18_O_2_	170.251	32.77
4	RT-15.974	000057-10-3	n-Hexadecanoic acid	C_16_H_32_O_2_	256.428	7.92
5	RT-17.430	000148-21-7	Vulgarin	C_15_H_20_O_4_	264.321	15.14

**Table 2 molecules-27-05188-t002:** Principle descriptors calculated for the selected ligands of *A. pallens*.

Compound Name	MW	HB Donor	HB Acceptor	SASA	QPlogPo/w	QPlogBB	QPlogS	%Human Oral Absorption
Lilac alcohol A	170.251	1	2.450	420.285	2.311	−0.030	−2.429	100
Spathulenol	220.354	1	0.750	451.722	3.937	0.355	−3.986	100
Lilac alcohol C	170.251	1	2.450	420.285	2.311	−0.030	−2.429	100
n-Hexadecanoic acid	256.428	1	2.000	675.898	5.282	−1.494	−5.593	87.129
Vulgarin	264.321	1	5.750	479.884	1.383	−0.616	−2.934	83.570

**Table 3 molecules-27-05188-t003:** SiteMap properties of MDR Ab-EP.

Binding Sites	AA Residues	S Score *	Size	D Score ^†^	Volume
Binding site_1	3,4,6,7,10,49,53,54,57,61, 65,97,100,101,103,104	0.844	55	0.861	219.520
Binding site_2	68,71,72,75,77,78,79,80,81, 82,90	0.742	43	0.666	128.968

* Site score. ^†^ Draggability score.

**Table 4 molecules-27-05188-t004:** Glide XP docking and MM/GBSA binding free energy (ΔG_bind_) results for ligands with target protein MDR Ab-EP.

Sl.No	Ligands	CAS Registry. NO	Amino Acid	H-Bond Interaction	Bond Length (Å)	Glide Score (kcal/mol)	MM-GBSA ΔG_bind_ (kcal/mol)
1	Lilac alcohol A	033081-34-4	TYR3 LEU6	H…O O…H	1.74 2.17	−3.706	−24.54
2	Spathulenol	006750-60-3	TYR3	H…O	2.08	−3.652	−30.51
3	Lilac alcohol C	033081-36-6	TYR3 LEU6	H…O O…H	1.74 2.17	−3.706	−24.54
4	n-Hexadecanoic acid	000057-10-3	ASN101	O…H	1.85	−2.187	−33.19
5	Vulgarin	000148-21-7	TYR3 TYR3	H…O O…H	2.05 2.13	−4.775	−39.34

**Table 5 molecules-27-05188-t005:** Average number of hydrogen bond interactions of MDR Ab-EP and ligand complexes at 50 ns.

Sl.No	Ligands	CAS Registry. NO	Average Hydrogen Bond Interactions
1	Lilac alcohol A	033081-34-4	0.02
2	Spathulenol	006750-60-3	0.01
3	Lilac alcohol C	033081-36-6	0.04
4	n-Hexadecanoic acid	000057-10-3	0.15
5	Vulgarin	000148-21-7	3.04

## Data Availability

Not applicable.

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
