# Peer review of "In Silico Evaluation of Bioactive Compounds of Artemisia pallens Targeting the Efflux Protein of Multidrug-Resistant Acinetobacter baumannii (LAC-4 Strain)"

_molecules, 2022, doi:10.3390/molecules27165188_

Round 1

Reviewer 1 Report

Suvaithenamudhan et al. have performed molecular docking and molecular dynamics simulations to evaluate the binding properties of five natural compound from A. pallens targeting the efflux protein of multi-drug resistant Acinetobacter baumannii (LAC-4 strain). They find that vulgarin and lilac alcohol A have a strong and stable binding efficiency with the efflux protein and could be considered as lead molecules for the discovery of new drugs against the multi-drug resistant Acinetobacter baumannii (LAC-4 strain) infection.

Some details of the modeling methods and analyses are missing, which weaken the reproducibility and robustness of their results and conclusion. Please see my comments below for details:

(1)   The MD simulations were only run once for each binding system, as well as for the free protein simulation. To increase the reproducibility and robustness of the results, the authors should run triplicate simulations for each system and study the average parameters across the three replicates.

(2)   The RMSD and RMSF analyses are both for the protein only. This can give some information of the structural fluctuation of the protein with the small molecule bound, but it cannot indicate the stability of the small molecule. The authors should compare the stability of the ligands (for example the RMSD of ligands vs. time); the stability of H-bonds (for example the probability of H-bond formation between protein and ligands vs. time) and the binding affinity (binding free energy) estimated by at least MM/GBSA or MM/PBSA level calculations.

(3)   Some important information of the MD simulation setup are missing and some are incorrect. At line 337, “NVT was also used for 100 ps to equilibrate the system with proteins and ligands at constant volume, pressure (1 atm), and temperature (300 K).”, the NVT ensemble doesn’t ensure the constant pressure. The ensemble information of the final MD simulations is not given. If the authors used NVT for the final MD simulations, there will be a problem of the system density and box size because it cannot be equilibrated in the NVT ensemble. NPT ensemble should be used first for density equilibration and then either NPT or NVT ensemble can be used for the final production run.

(4)   The amino acid sequence of the modeled efflux protein or its accession number of a database must be given.

(5)   In Table 5, what is “average hydrogen bond interactions”? Also in the line 212, “The maximum number of hydrogen bond interactions is shown in Table 5.”, the authors used the term “maximum number of hydrogen bond interactions”, which is not consistent with Table 5. The details of this analysis must be given in the Method section.

Other comments:

(1)   In Abstract, “GC-MS analysis of the ethanol extract of A. pallens detected four major compounds…”. However, there are 5 compounds listed.

(2)   “ADMET” should give the full name at the first time it is mentioned.

(3)   Line 78, “…spectroscopic (FTIR) analysis…”. The full name of FTIR should be given.

(4)   Figure 1, the X-axis and Y-axis labels and units are missing.

(5)   The sentence at line 213 to line 216 seems misplaced. It looks like a conclusion of the previous section.

(6)   Fig. 7B doesn’t have a Y-axis label. In the caption, “Bar diagram indicating the average of RMSD values…”, were these values averaged by using the entire trajectories or the equilibrated section?

(7)   Line 296, “…was assigned using OPLS3e force field method…”. The force field is not a “method”. Also, the citation for this force field is needed.

(8)   Line 300, missing citation for “Chemistry Web Book”.

(9)   Line 305, missing citation for “OPLS4 force field”. Also, why did the authors use OPLS3e for ligand and OPLS4 for protein? 

Reviewer 2 Report

1. In the introduction section author should provide the mortality rate caused by Acinetobacter baumannii infection

2. In the introduction section author should clearly state why they chose/targeted particularly the bioactive compounds of Artemisia pallens. 

3. The author should suggest some wet laboratory-based assays in the conclusion section for future study. 

Author Response

Reply to the Reviewer 2:

We thank the reviewer for his valuable suggestions.  We have carried out all the suggestions mentioned by the reviewer. 

  1. In the introduction section author should provide the mortality rate caused by Acinetobacterbaumannii infection

Reply:

  • We have provided the morality rate caused by Acinetobacterbaumannii infection as follows:

Nosocomialpneumonia acquired due to A. baumannii was reported with a mortality rate of 30 to 75% while, community-acquired pneumonia (CAP) due to A. baumannii had a mortality rate of 40 to 60% [4]. Recently, SENTRY antimicrobial surveillance program identified high prevalence of MDR - A. baumannii in many parts of the world including India, Pakistan, Malaysia, Thailand and Taiwan [5-7]”.

The above lines have been included in the ‘Introduction’ section.  (Please See Line numbers: 46 - 51).

We have also included the relevant citations [4-7] and the details are included in the ‘References’ section. (Please See Line numbers:  397-404).

  1. In the introduction section author should clearly state why they chose/targeted particularly the bioactive compounds of Artemisia pallens

Reply:

  • The reason for selecting these bioactive compounds is mainly due to the following reason.

“Presence of vast variety of secondary metabolites makes this plant an ideal candidate to be used as herbal therapeuticagainst various ailments”.

The above line has been included in the ‘Introduction’ section.  (Please See Line numbers :80 - 82).

  1. The author should suggest some wet laboratory-based assays in the conclusion section for future study. 

Reply:

For the “wet laboratory based assays for future studies”, the following lines have been added in the ‘Conclusion’ section. (Please see Line numbers :375 - 379).

 “The antimicrobial activity of the reported compounds should be validated through qualitative estimation such as agar diffusion assay and minimal inhibitory concentration (MIC) needs to be carried out for future consideration of the bioactive compounds as an antibacterial agent. With these considerations,”.

With these modifications, we hope that our revised manuscript may be acceptable for publication.

Round 2

Reviewer 1 Report

The authors have addressed most of my comments and the manuscript has been improved.

However, my major concern #3 is not fully addressed in this revision. It is not correct to say "NVT was also used for 100 ps to equilibrate the system with proteins and ligands at constant volume, pressure (1 atm), and temperature (300 K)" because the NVT essemble ensures constant volume and temperature but not pressure. The added sentence "In a similar way, NPT ensemble was used for density equilibration" is confusing because the order of these two equiliration simulations is unknown. Moreover, the essemble used in the final 50 ns simulations is still not provided. Missing such information significantly decreases the reproducibility of this study. I suggest the authors carefully rewrite method section 3.9 to make the simulation setups and steps clear to readers.

This manuscript is suitable to be accepted for publication after the minor revision.

Author Response

Reply to the Reviewer’s comment:

The authors have addressed most of my comments and the manuscript has been improved.

 However, my major concern #3 is not fully addressed in this revision. It is not correct to say "NVT was also used for 100 ps to equilibrate the system with proteins and ligands at constant volume, pressure (1 atm), and temperature (300 K)" because the NVT essemble ensures constant volume and temperature but not pressure. The added sentence "In a similar way, NPT ensemble was used for density equilibration" is confusing because the order of these two equiliration simulations is unknown. Moreover, the essemble used in the final 50 ns simulations is still not provided. Missing such information significantly decreases the reproducibility of this study. I suggest the authors carefully rewrite method section 3.9 to make the simulation setups and steps clear to readers.

 This manuscript is suitable to be accepted for publication after the minor revision.

Reply:

We thank the reviewer for mentioning that the manuscript has witnessed improvement. 

As suggested by the reviewer, we have carefully rewritten the method Section 3.9 to make the simulation setups and steps clear to readers. We have made the following modifications in the revised manuscript in Section 3.9.

“The equilibration of the protein-ligand complexes were conducted in two phases; the first conducted under an NVT ensemble (Constant Temperature and Constant Volume) for 100 ps to equilibrate the system with proteins and ligands at constant volume, and temperature (300 K), and the second phase under NPT (Constant Temperature and Constant Pressure) ensemble, wherein the number of particles, pressure, and temperature are kept constant. NPT was also used for 100 ps to equilibrate the system with proteins and ligands at constant volume, pressure (1 bar), and temperature (300 K). After completion of the two equilibration phases, the systems were well-equilibrated at the desired temperature and pressure for MD analyses.”(Please see the Line numbers 357-365 in the revised manuscript)

We have also carried out the English language editing and spell check in the revised manuscript.

Therefore, we hope that our revised manuscript is acceptable for publication.